# Student Pharmacists during the Pandemic: Development of a COVID-19 Knowledge, Attitudes, and Practices (COVKAP) Survey

**DOI:** 10.3390/pharmacy9040159

**Published:** 2021-09-30

**Authors:** Alina Cernasev, Meghana Desai, Lauren J. Jonkman, Sharon E. Connor, Nicholas Ware, M. Chandra Sekar, Jon C. Schommer

**Affiliations:** 1Department of Clinical Pharmacy and Translational Science, College of Pharmacy, University of Tennessee Health Science Center, 301 S. Perimeter Park Dr., Suite 220, Nashville, TN 37211, USA; 2Health Analytics Network, LLC, Pittsburgh, PA 15237, USA; mdesai@healthanalytics.net; 3Department of Pharmacy and Therapeutics, University of Pittsburgh School of Pharmacy, 5607 Baum Blvd, Suite 303, Pittsburgh, PA 15206, USA; ljf1@pitt.edu (L.J.J.); sconnor@pitt.edu (S.E.C.); 4Department of Clinical Pharmacy and Translational Science, College of Pharmacy, University of Tennessee Health Science Center, Memphis, TN 38163, USA; nware@uthsc.edu; 5College of Pharmacy, University of Findlay, Findlay, OH 45840, USA; sekar@findlay.edu; 6Department of Pharmaceutical Care & Health Systems, College of Pharmacy, University of Minnesota, 308 Harvard Street SE, Minneapolis, MN 55455, USA; schom010@umn.edu

**Keywords:** student pharmacist, COVID-19 pandemic, remote learning

## Abstract

Background: The COVID-19 pandemic has caused innumerable changes to all aspects of human life and behavior, including academic life. This study describes the development of a COVID-19 Knowledge, Attitudes, and Practices (COVKAP) Survey among U.S. student pharmacists. The survey was administered at Doctor of Pharmacy programs in three states—Tennessee, Ohio, and Pennsylvania. Methods: The COVKAP survey—an online cross-sectional survey—was distributed to U.S. student pharmacists enrolled in three different colleges of pharmacy in three states during the fall semester of 2020. The survey was developed using literature review and Dillman’s recommendations for survey design. The COVKAP survey consisted of 23 closed and Likert-scale questions, and three open-ended questions. The research team conducted descriptive and inductive thematic analyses on the quantitative and qualitative data, respectively using SPSS (v27) and Dedoose^®^ software. Results: A total of 421 responses were received. Respondents were predominantly female (72%) and White (79%). The average age of respondents was 23.4 years. The qualitative analysis revealed three themes: (1) Wellbeing and mental health struggles; (2) Being part of the decision-making process; (3) Necessity of adequate protection measures. Conclusions: Preliminary study findings indicate that student pharmacists’ concerns and the challenges that they face in their academic pursuits are largely similar across the three states in this study and inform about the importance of recognizing and mitigating the impact of widespread disruption in education. This disruption provides an opportunity for pharmacy academia to examine practices and methods that can be improved upon to help students become successful practitioners.

## 1. Introduction

The COVID-19 pandemic has caused innumerable changes to all aspects of human life and behavior, not least of which is the education system [1]. Now, more than a full year since the declaration of the pandemic in March 2020, behaviors such as social distancing and mask-wearing have been incorporated into daily lives [2]. The full impact of the pandemic on educational and academic outcomes will likely unravel in coming years. In response to the pandemic-related disruption, various schools, colleges, and universities of higher education across the U.S. initiated a series of actions such as remote learning platforms, limiting class sizes, minimizing in-person interactions, enforcing mask-wearing on campus, and limiting visitors [3,4]. Notably, for many educational institutions, such measures are new and there has been little time to adapt to teaching and learning in alternate settings. Further, actions have varied by state and type of education provider (public vs. private), leading to a lack of uniformity in education provision amidst the severe and sudden disruption.

In response, students needed to adapt to online learning, living in relative isolation, and being fearful for their own and their loved ones’ safety [3,5,6]. The profound impact of such disruptions on academic achievement and professional development in the long term remains to be seen. In particular, health science students experienced additional complications due to the disruption in their continued public and clinical responsibilities as budding health professionals, compounded by the additional needs and challenges in balancing personal safety with public responsibility [6,7,8]—including experiential learning and students’ ability to interact in-person with patients, and with each other [9,10]. The rapid switch to remote or electronic learning compounded the burdens of stress experienced by health students [9,11]. Early literature on academic disruption indicates that students indicated frustration with technical difficulties in the delivery and modes of instruction [8], and conveyed challenges in their ability to concentrate, notwithstanding the abrupt loss of face-to-face interactions that provide meaningful dialogue and enhanced learning for many students [11].

Beyond didactic instruction, the pandemic has a yet-to-be-measured impact on clinical training. In a study of health professions students including pharmacy students, perceptions of readiness for clinical experiences worsened [12]. A study of medical students [13] reported that almost three-quarters of upper-level students felt that the pandemic significantly disrupted their clinical training, and reported mixed feelings of guilt, relief, and disappointment at the beginning of the pandemic, along with concerns both about their own safety [12], and of being underutilized by health systems [8], possibly creating a sequelae of demotivation and consequences for educational and skill-based outcomes in the longer term.

It is important to note here the downstream impact of the pandemic on the mental health of students, including accentuating and creating stressors (fear and worry for oneself or loved ones), constraints on physical movement and social activities due to quarantines, and sudden and radical lifestyle changes, in addition to the loss of economic stability for many students and their families [7,14]. Isolation due to quarantines may have a deeper impact on those already suffering from mental health disorders and challenges. Early reports of student perceptions of their wellbeing and connectedness to their classmates and the faculty have varied [8]. In the short term, there is emerging evidence of trauma caused due to the pandemic; however, the potential long-term effects on students, their educational outcomes and their mental health and wellbeing, are yet to be revealed [14].

Student pharmacists lost their opportunity to gain valuable clinical insights during their Advanced Pharmacy Practice Experiences (APPE) and experienced changes in experiential rotation formats, along with difficulties meeting licensure requirements for those who were graduating in 2020 and inability to serve alongside licensed pharmacists, missing out on the accompanying sense of accomplishment and pride. Student pharmacists play an important role in the healthcare system [10], and there are reports of participation and service provided by student pharmacists during the pandemic [15]. While many student pharmacists have adapted and adjusted to continue their academic pursuit, their knowledge of the COVID-19 pandemic, attitudes towards the rapidly changing environment, and their practices during the pandemic are lesser known. As the pandemic continues in 2021 and possibly beyond, it is important to capture the rapidly evolving situation in terms of adaptations and behaviors to inform pharmacy academia. Thus, the purpose of this study was to characterize pharmacy students’ knowledge, attitudes, and practices related to the COVID-19 pandemic through the development of a multi-item survey and administer it to a small subset of the student pharmacist population.

## 2. Methods

The COVKAP study is a cross-sectional survey developed by a team of researchers using Dillman’s recommendations for survey design and a review of literature on COVID-19 [16]. The COVKAP survey consists of three open-ended questions and 37 multiple choice questions (12 questions about COVID-19 knowledge, 15 on COVID-19-related attitudes, 5 on COVID-19-related practices, and 5 questions on returning to in-person instruction) using a Likert scale, categorical, and fill-in-the-blank response options (See Appendix A for the COVKAP survey). The open-ended questions explore student pharmacists’ experiences of remote/virtual learning and its implications. The quantitative section focuses on demographics, opinions about transmission, treatment of COVID-19, and opinions about the impact of COVID-19 on their experiences as student pharmacists. At the beginning of the survey, participants answer questions regarding demographics and pharmacy work history. Work history questions included the setting and duration of employment. The survey participant could choose not to respond to any questions within the survey. The study was approved by the Institutional Review Boards (IRB 20-0756-XM) of the University of Tennessee Health Science Center (UTHSC, approved on 11 August 2020), University of Pittsburgh (IRB 20090154) (Pittsburgh, approved on 6 September 2020), and University of Findlay (IRB 1482) (Findlay, approved on 9 October 2020).

Student pharmacists enrolled in the Doctor of Pharmacy program at UTHSC, Pittsburgh, and Findlay voluntarily completed the electronically delivered COVKAP survey. To accommodate the differences between starting months for each institution, the survey was administered over three months in the fall of 2020, with monthly reminders for completion. Inclusion criteria were student pharmacists enrolled in the Pharm.D. program at the time of survey administration.

Qualitative section: The survey finishes with three open-ended questions that ask about student pharmacists’ concerns regarding resuming in-person instruction. An inductive thematic analysis was conducted [17] to procure direct information from the respondents without imposing defined categories or theories. One researcher (AC) read the comments several times, imported the text into Dedoose^®^ (Manhattan Beach, CA, USA), a qualitative analysis software, and analyzed the data. The extracted codes were grouped on similarities that resulted into categories, and subsequent themes. To test the reliability of the codes, the second researcher (JS) read each code, descriptors, the relevant categories, and themes [17]. The two researchers discussed the codes and categories, arbitrated differences, and clarified themes [17].

Quantitative data: Descriptive statistics were used to summarize the pattern of findings using SPSS (v27) (https://www.ibm.com/support/pages/node/3006603).

## 3. Results: Quantitative Results

A total of 421 student pharmacists completed the COVKAP survey—Pittsburgh (98), Findlay (106), and UTHSC (217), respectively. Response rates were 65% for Pittsburgh and Findlay, and 31% for UTHSC. The average age of respondents was 23.4 (±3.4) years, with a minimum of 18 years and a maximum of 66 years. Respondents were predominantly female (72%) and White (79%).

Of the respondents, 34% (143) were currently working as interns and 62% (261) reported that they had worked in a pharmacy setting prior to pharmacy school. At the time of survey completion, 36% (151) respondents were engaged in online instruction only. 46% (193) of the respondents reported that they lived in a household of two or fewer people (including themselves) and only 6% (25) lived in a household with a person of age 65 or older. See Table 1 for student pharmacists’ demographics.

When asked about whose responsibility it would be to decide about returning to school, 61% (256) reported “solely or mostly mine,” 24% (101) “solely or mostly the school,” and 15% (63) responded “mine and my family.” Regarding preferences for resuming classes, 32% (134) preferred “Hybrid in-class and online instruction,” 30% (126) preferred “In-class instruction with appropriate social distancing,” 26% (109) preferred “Online instruction only,” and 12% (50) preferred “In-class instruction with no social distancing.”

In terms of student perceptions, results in Figure 1 indicate that only 15% either agreed or strongly agreed (SA) that COVID-19 was under control in the U.S. during Fall 2020. Five percent strongly agreed that PPE was available to all healthcare workers, and 11% strongly agreed that PPE was available to pharmacists. However, it is interesting to note that only 35% strongly agreed to getting vaccinated. The impact of mask-wearing or not wearing by patients appear to be relatively the same (27% SA for negative interactions vs. 26% for positive interactions).

The analyzed qualitative data emerged into three themes: (1) Wellbeing and mental health struggles; (2) Being part of the decision-making process; (3) Necessity of adequate protection measures.

The first theme focused on the student’s wellbeing and described the students’ mental health struggles for daily life activities. Most of the participants found that the online classes did not offer them enough breaks. Several students commented on personal experiences that had contributed to the development of isolation and mental health issues. Other participants described challenges such as lectures to be recorded, no additional assignments and family addressing their own medical needs. For example, at PITT, students went from the beginning of the semester to finals without a single day off. Participants described their frustrations with the large number of exams that contributed to their enhanced stress levels. The representative quotes are in Table 2.

In the second theme, the participants highlighted the need to be part of the decision-making process for resuming in-class instruction. For example, the participants described the process of decision-making as an essential part of their daily activities, and they felt that the university should ask for their opinion before returning to in-class learning activities. Furthermore, several participants mentioned challenging decisions, such as returning to in-class lectures when they might have medical conditions that prevent them. The representative quotes are located Table 3.

The third theme presents the participants’ opinion on the adequate protection measures necessary for a smoother transition when resuming in-class instruction. For example, many participants presented their views about social distancing usage and how it could be enforced even when the classroom space is limited. Additionally, a few students described the importance of the availability of free cleaning supplies in the university. The representative quotations for the third theme can be found in Table 4.

## 4. Discussion

The findings of this study provide interesting insights into student pharmacists’ stresses due to the disruptions caused by the COVID-19 pandemic. A novel finding of this study is the first theme, which provides additional information about pharmacy students’ mental health issues representing a barrier to their wellbeing. It is important to note that many respondents commented about stress and change in sleep patterns without the research team prompting in the survey. While few studies have mentioned the stress level of the pandemic and coping mechanisms among healthcare students [7,18], pharmacy education needs to consider the mental and psychological needs of student pharmacists.

Interestingly, most students disagreed that COVID-19 was under control in the U.S. during Fall 2020, and perceived that PPE availability was low for healthcare workers and pharmacists. More than half of the survey respondents did not feel adequately prepared to administer the COVID-19 vaccine, and about 61% agreed to get vaccinated. An interesting finding in this study is the contrast between perceived risk and proposed behaviors. Mask-wearing by patients did not comfort or discomfort students. Over 75% student pharmacists felt that COVID-19 questions should be directed to physicians and not pharmacists, yet 55% indicated willingness to counsel their own family members. Considering the widespread vaccine hesitancy, different PPE mandates, and varying regulations across States, these findings provide an insight into this unique population.

Other unique findings in this study include the need for fall or spring breaks and longer breaks between online lectures, which would enable student pharmacists to disconnect. These insights could be helpful to pharmacy educators when designing lectures and planning for breaks within lectures. For example, Findlay respondents recommended all lectures be recorded. On the other hand, students at Tennessee and Pittsburgh were satisfied with recorded lectures.

Indeed, growing concern about stress experienced by student pharmacists may have detrimental effects on overall wellbeing [19,20]. In a university-wide study on mental health status and the severity of depression and anxiety during the current pandemic [9], it was revealed that 71% of students experienced enhanced stress/anxiety levels during the pandemic [9]. The decision-making process results could be useful for educators and universities when making decisions regarding returning to in-person classes [21,22]. Study responses indicated student pharmacists’ desire to be a part of the decision-making process for returning to in-person lectures. It is also important to note that opinions about preferred learning modality were split evenly amongst respondents, showing a potential for conflict regarding continued online learning compared to a return to in-person class. Students discussed factors from both an individualistic (what is best for me) and a collectivistic (what is best for my loved ones or what is best for society) perspective. Factoring in student preferences along with measurement of educational achievement and outcomes could pave the way for reshaping how formal education is imparted to student pharmacists. Indeed, the disruption in routines necessitates critical thinking about new modes of blended/hybrid learning that could lead to better engagement and improve instruction in pharmacy education.

The results from this study inform educators and administrators about student pharmacists’ preferences for classes and developing curricula for pharmacy education to help meet their needs, which may be explored beyond the pandemic. It may be noted that few pharmacy programs have implemented different strategies to help students cope with stress levels [14]; however, widespread adoption of formal measures remains to be seen. Pharmacy educators are well positioned to address student concerns and explore opportunities to reduce stress and optimize the learning environment. Leadership in pharmacy academia may consider exploring pathways to further enhance and adopt evidence-based strategies to address mental health and wellbeing of their students toward developing resiliency and preparing themselves, their educators, and student pharmacists for future disruptions.

## 5. Limitations

This exploratory study describes the COVKAP survey development to capture student attitudes, perspectives, and behaviors during the pandemic. Therefore, study results should be considered in light of some limitations. Firstly, each university started their Fall 2020 semester at different times and in different modes (online only vs. in-person vs. hybrid) due to pandemic disruptions, and survey responses may be influenced by university and state responses to the pandemic. The results of this study may be limited to similar geographic and demographic characteristics. Secondly, this study had a limited number of responses, which could be due to pandemic disruptions and student pharmacist burnout. Further, this study included students across all years in the professional program, so the impact of the pandemic may be different for fourth year students on APPEs compared to first professional year students who were learning in a new program in the midst of a global pandemic.

Lastly, we described preliminary findings from the COVKAP survey, and further studies are underway with additional student populations to conduct further statistical analyses.

## 6. Conclusions

Student pharmacists’ experienced severe disruption in their academic journey. Yet, they have adapted, adjusted, and made necessary changes as they prepared to resume in-person instruction and their gradual return to campus life. As the pandemic continues in 2021, it is time to evaluate and value student pharmacists’ experiences as future healthcare practitioners. Investments into their educational experiences can yield richer outcomes in terms of high-quality practitioners. Challenges around mental health have existed for many years, yet little has been done to improve and/or address students’ mental health needs in a systematic, evidence-based manner. This disruption provides an opportunity for leadership in pharmacy academia to examine practices and strategies that can be improved upon to help students develop resilience and become successful practitioners. Indeed, as our preliminary study indicates, there is an urgent need to address critical knowledge gaps, develop scientific temper, and promote resiliency building among student pharmacists.

## Figures and Tables

**Figure 1 pharmacy-09-00159-f001:**
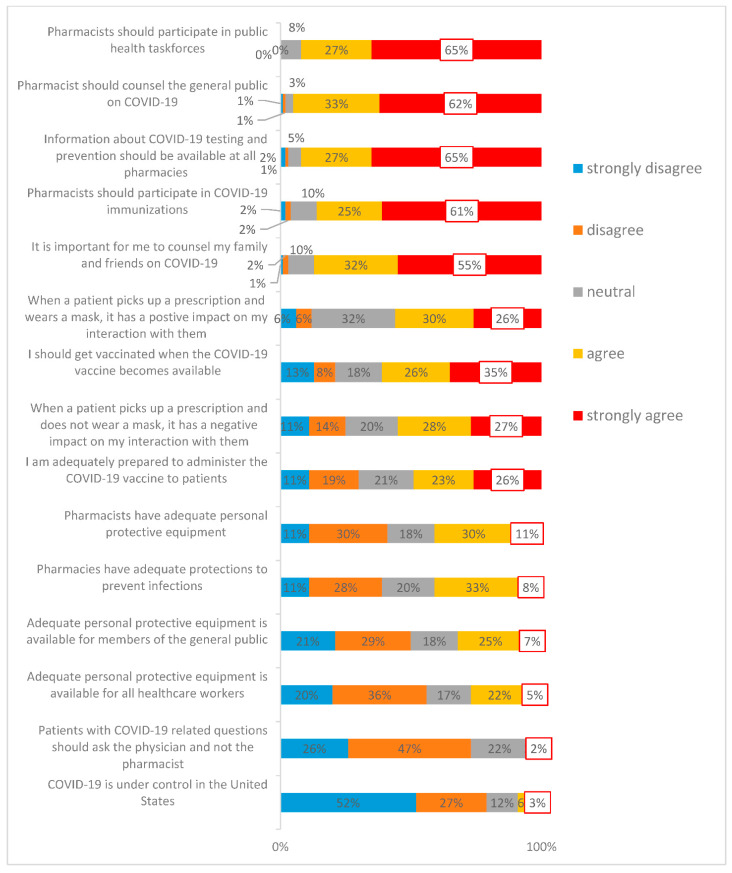
Opinions about the impact of COVID-19 on student pharmacist experiences.

**Table 1 pharmacy-09-00159-t001:** Demographics characteristics of the survey sample.

Demographics	Tennessee	Pittsburgh	Findlay
**Age** (mean, SD)	24 ± 3	23 ± 4	22 ± 1
**Gender** (%)			
Male	28%	25%	27%
Female	72%	72%	73%
Other	0%	3%	0%
**Race** (%)			
White	79%	70.0%	90%
Black	7%	2%	4%
Asian	15%	24%	6%
Other	0%	4.0%	0.9%
**Mode of Instruction** (%)			
Online only	60%	48%	90%
In-person or commute to campus	41%	52%	10%
**Year of graduation** (%)			
2021	14.0%	23%	24%
2022	19%	34.0%	35%
2023	26%	13%	10%
2024	41%	31%	30%

**Table 2 pharmacy-09-00159-t002:** Supportive quotes for Theme 1: Wellbeing and mental health struggles.

Supportive Quotes
“We are socially isolated. Try to create virtual event where we can meet other students, that we actually want to go to.”
“The fall semester took a bit toll on my mental health and I’m sure I’m not the only one who feels that way. It would be nice to get some breaks in between instead of making just one long winter break like we did. I would also like more events that promote mental well-being because I think a lot of people need support right now.”
“Although it may seem like we have more time, the workload is still massive and actually is too much for online classes. We have other classes too and students are also expected to work more because classes are online.”
“I understand it is hard for teachers to prepare for where we are now, but I would appreciate more appropriate learning techniques for the online setting. Getting our attention is not that hard. I would also like to see more attention to the mental health of the student.”
“The university needs to be aware of the mental toll that an accelerated all online semester has on students. I’d ask that faculty still be aware of how online instruction and a global pandemic could be impacting students’ health. Spending the majority of the day behind a computer either in lecture or doing work outside of class isn’t the best thing for students’ health. I would suggest having a guideline for breaks during long classes so that students can give their eyes a break and move around during 3+ hour lectures.”
“I need to have time away from my computer and have teachers not run over on time. Just because I am home, does not mean that I am not busy.”
“The constant stress with no break has negatively impacted all the students. it has led to constant anxiety and poor performance on exams because students haven’t had time to relax after constant exams.”
“Consider the burnout students are experiencing about this point in the semester because our fall break got taken away.”

**Table 3 pharmacy-09-00159-t003:** Supportive quotes for Theme 2: Being part of the decision-making process.

Supportive Quotes
“It is personal choice of the student to be in person. “
“Life is more important than money. People are dying. We should not be in-person in class.”
“Even if I am not high risk or living with high risk individuals, I should be able to participate online.”
“If there is an in-person option, I would prefer to be asked before I return to school.”
“Commuting in winter is more difficult and makes it more likely that I will have to use public transport which I avoided last semester if classes are only in-person.”
“The effect returning to classes without appropriate social distancing would have on immunocompromised individuals. Ask us before making a decision.”
“I would prefer most lecture classes be online and labs where working in groups is necessary or working on cases, etc. to be in person.”
“No pressure of returning to in-person type activities.”
“Students need person versus person interaction and the university should consider the students voice the most even if that means they want to put themselves more at risk.”

**Table 4 pharmacy-09-00159-t004:** Supportive quotes for Theme 3: Necessity of adequate protection measures.

Supportive Quotes
“Strict social distancing enforcement. Enforcement of masks on campus. Close public buildings and have better isolation and quarantine practices.”
“Students comfort level of coming into a crowded classroom.”
“Social distancing should be in place. Masks should be enforced.”
“Currently, I feel like everyone has their own definitions of what is proper social distancing behavior because there has not been a unified voice saying what is acceptable in terms of any social activities. I think the schools need to be realistic in terms of setting rules for students who want to socialize with each other safely.”
“Proper social distancing. Random testing (net just those who show symptoms.”
“ability to provide PPE to students, ability to provide adequate sanitizing of the class areas.”
“We aren’t protected enough, high touch surfaces are not cleaned enough, we do not have the ability to social distance in class, their making us risk our health and family members health.”
“More wipes present. more hand sanitizer available.”
“Continue to provide masks, sanitizer, disinfecting wipes. Continue tracking.

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
