# Peer review of "Student Pharmacists during the Pandemic: Development of a COVID-19 Knowledge, Attitudes, and Practices (COVKAP) Survey"

_pharmacy, 2021, doi:10.3390/pharmacy9040159_

Round 1

Reviewer 1 Report

This paper described a survey study regarding student experiences during COVID-19. The following comments are offered in an effort to help strengthen the work:

  1. Please add a purpose statement or research questions at the end of the introduction.
  2. Please provide the response rate for each school and overall
  3. The methods are missing a description of how the survey was validated - were any psychometrics conducted to demonstrate the validity of your data?
  4. Please add text to the results, and a callout, for the data in Figure 1.
  5. Please describe in the results what the data in Figure 1 mean, and how they relate to the other data collected.
  6. Please adjust numbers in the tables to be consistent with points after the decimal (ie some have 0, some have 2, some have 3)

Author Response

  1. Please add a purpose statement or research questions at the end of the introduction. Response: Thank you for the valuable recommendation. The text was amended.
  2. Please provide the response rate for each school and overall. We value your suggestion. Response:The response rate was added to the results section.
  3. The methods are missing a description of how the survey was validated - were any psychometrics conducted to demonstrate the validity of your data? Thank you for this recommendation. Response: The methods part describes how the survey was developed.
  4. Please add text to the results, and a callout, for the data in Figure 1. Response: We value your recommendation. The results section was amended.
  5. Please describe in the results what the data in Figure 1 mean, and how they relate to the other data collected. Response:We value your recommendation. The results section was amended.
  6. Please adjust numbers in the tables to be consistent with points after the decimal (ie some have 0, some have 2, some have 3) Response:  We value your recommendation. We made the changes.

Reviewer 2 Report

Dear Authors,

Thank you for the opportunity to read and review this paper. Although it's an interesting text, there are some points which might be improved in my opinion. Please find my suggestions below:

  1. Please consider changing the title. The study is somewhat preliminary. It's good to point it at the beginning.
  2. In my opinion, the paper would benefit if the introduction will be shorter. E.g., the third paragraph seems to be not relevant. Besides, much information in this section is well known. These were global problems.
  3. Please clearly present the aim of the study in the main body file.
  4. Would you mind presenting inclusion and exclusion criteria within the methods section?
  5. A small study group is a limitation of that study. If I were you, I would point it out in the Limitation section.

Author Response

  1. Please consider changing the title. The study is somewhat preliminary. It's good to point it at the beginning. Response: Thank you for your suggestion. However, we believe that the word “development” captures the preliminary nature of this study.
  2. In my opinion, the paper would benefit if the introduction will be shorter. E.g., the third paragraph seems to be not relevant. Besides, much information in this section is well known. These were global problems. Response: We value your recommendation. The introduction was amended.
  • Please clearly present the aim of the study in the main body file. Response: Thank you for the valuable recommendation. The text was amended.
  • Would you mind presenting inclusion and exclusion criteria within the methods section? Response: Thank you for this suggestion. We added the inclusion and exclusion criteria.
  • A small study group is a limitation of that study. If I were you, I would point it out in the Limitation section. Response: We value your recommendation that was incorporated into the limitation section.

Round 2

Reviewer 2 Report

Thank you for your answers. In my opinion, the study improved and it might be considered for publication in a present form.